# Globally consistent estimates of high-resolution Antarctic ice mass balance and spatially-resolved glacial isostatic adjustment

Matthias O. Willen[1,a], Martin Horwath[1], Eric Buchta[1], Mirko Scheinert[1], Veit Helm[2], Bernd Uebbing[3], and Jürgen Kusche[3]

[1]Institut für Planetare Geodäsie, Technische Universität Dresden, Germany
[2]Alfred Wegener Institute, Helmholtz Centre for Polar and Marine Research, Bremerhaven, Germany
[3]Institute of Geodesy and Geoinformation, University of Bonn, Germany
[a]now at: Department of Geoscience and Remote Sensing, Delft University of Technology, The Netherlands

**Correspondence:** Matthias O. Willen (matthias.willen@tu-dresden.de)

**Abstract**: A detailed understanding of how the Antarctic Ice Sheet (AIS) responds to a warming climate is needed because it will most likely increase the rate of global mean sea level rise. Time-variable satellite gravimetry, realized by the Gravity Recovery and Climate Experiment (GRACE) and GRACE-Follow-On (GRACE-FO) missions, is directly sensitive to AIS mass changes. However, gravimetric mass balances are subject to two major limitations: First, the usual correction of the glacial isostatic adjustment (GIA) effect by modelling results is a dominant source of uncertainty. Second, satellite gravimetry allows for a resolution of a few hundred kilometres only which is insufficient to thoroughly explore causes of AIS imbalance. We have overcome both limitations by the first global inversion of data from GRACE/GRACE-FO, satellite altimetry (CryoSat-2), regional climate modelling (RACMO2), and firn densification modelling (IMAU-FDM). The inversion spatially resolves GIA in Antarctica independently from GIA modelling jointly with changes of ice mass and firn air content at $50\,\mathrm{km}$ resolution. We find an AIS mass balance of $-144 \pm 27\,\mathrm{Gt\,a^{-1}}$ from Jan 2011 to Dec 2020. This estimate is the same, within uncertainties, as the statistical analysis of 23 different mass balances evaluated in the Ice sheet Mass Balance Inter-comparison Exercise (IMBIE, Otosaka et al., 2023b). The co-estimated GIA corresponds to an integrated mass effect of $86 \pm 21\,\mathrm{Gt\,a^{-1}}$ over Antarctica and it fits better with GNSS results than other GIA predictions. From propagating covariances to integrals, we find a correlation coefficient of $-0.97$ between the AIS mass balance and the GIA estimate. Sensitivity tests with alternative input data sets lead to results within assessed uncertainties.

## 1 Introduction

Satellite-based estimates of the mass balance of the Antarctic Ice Sheet (AIS) allow one to quantify the response of the AIS to global warming. Projections show that solely the AIS may contribute between $4\,\mathrm{cm}$ to $34\,\mathrm{cm}$ to global mean sea level until 2100 (Fox-Kemper et al., 2021a). However, even recent mass loss estimates of the AIS vary over a wide range, e.g. from 94 to $202\,\mathrm{Gt\,a^{-1}}$ over the time period 2010–2019 (Fox-Kemper et al.,

2021b). This large spread reveals a lack of knowledge which propagates to projections.

The mass balance of an ice sheet—also referred to as the ice mass change (IMC) of an ice sheet—is the difference of the input mass flux, i.e. mainly the accumulation by precipitation, and the output flux, i.e. for the largest part ice discharge and meltwater runoff into the ocean. Commonly, three methods are applied to determine ice sheet's mass balance using satellite data: (i) the gravimetric method deriving the mass balance from gravitational field changes measured by GRACE and GRACE-FO satellite missions, (ii) the altimetric method deriving the mass balance from surface elevation changes measured by several radar and laser altimeter missions while assuming a volume to mass conversion, and (iii) the mass budget method deriving the mass balance by assessing the difference between the input and output mass fluxes. They are derived from regional climate modelling and from ice discharge estimates which can be retrieved from remote-sensing satellite data and ice thickness data. All three methods have advantages but also limitations, which contribute to the large spread of estimates mentioned above, extensively documented elsewhere (e.g., Otosaka et al., 2023b; Otosaka et al., 2023a). To summarize, (i) has the advantage that it is directly sensitive towards mass changes but there is the need to exclude all other sources of mass redistributions, which are superimposed in the gravitational field changes. The present-day effect due to glacial isostatic adjustment (GIA) is the most relevant and most uncertain (e.g., Groh and Horwath, 2021). The term present-day GIA effect refers to the presently observable effects resulting from the adjustment process to an isostatic state, which was induced by glacial mass changes in the past. This is to be distinguished from effects associated with the instantaneous elastic response to con-temporaneous ice-mass loading changes (Thomas et al., 2011). GIA predictions for Antarctica differ by several tens of gigatons per year and disagree in their spatial patterns due to assumption on rheology and ice loading history (Whitehouse et al., 2019). Furthermore, IMC estimates derived from gravitational field changes, usu-ally, only allow for a spatial resolution of a few hundred kilometres. (ii) has the advantage to capture IMC with high spatial resolution (e.g., Schröder et al., 2019) but the conversion from volume changes to mass changes is based on effective density hypotheses or needs to include auxiliary data, e.g. firn modelling results where the uncertainties are largely unknown. (iii) has the advantage that it aims to resolve the full mass fluxes and not only the differential signal between input and output fluxes (Rignot et al., 2019). However, as the mass balance amounts only less than 10 % of the magnitude of the mass fluxes, even small errors have strong impact on the result. The input flux and the output flux are each subject to an uncertainty that is in the order of magnitude of the AIS mass balance itself (Mottram et al., 2021).

In addition to these mass balance estimation strategies, there are methods that combine data from satellite gravimetry and satellite altimetry to build on the advantages of both sensors. Some of the combination ap-proaches aim to co-estimate the GIA effect rather than using GIA modelling results to account for the GIA signal (Wahr et al., 2000; Riva et al., 2009; Gunter et al., 2014; Martín-Español et al., 2016; Sasgen et al., 2017; Zhang et al., 2017; Engels et al., 2018; Willen et al., 2020; Zwally et al., 2021). These regional approaches implemented regional constraints to evaluate the global gravitational fields from satellite gravimetry in a re-

gional domain. Except for Martín-Español et al. (2016), these approaches only allow a smoothed estimation of IMC with a spatial resolution comparable to GRACE/GRACE-FO-only estimates. In addition to these regional approaches, there are global inversion approaches that co-estimate the GIA signal (Rietbroek et al., 2016; Jiang et al., 2021) by fitting prescribed GIA spatial patterns to data or utilizing the signal co-variance information. These approaches are limited in the sense the estimated GIA depends on the applied modelling output. Hence, they fit presumably erroneous a priori information to the input data. The approaches according to Rietbroek et al. (2016) and Jiang et al. (2021) only allow for basin-wise and a smoothed estimation of AIS IMC, respectively.

Here we extend the work of Willen et al. (2022) by applying a global inversion framework with a focus on Antarctica. We use data sets from satellite gravimetry, satellite altimetry, and regional climate and firn modelling. The latter are used to derive changes of the Firn Air Content (FAC) based on the surface mass balance (SMB) from a regional climate model and firn thickness changes of a firn densification model (FDM). This approach aims to overcome some limitations of previous combination approaches that allow to jointly determine IMC and GIA. First, it is a global framework, i.e. no regional constraints need to be implemented, and mass changes are parametrized across the globe. Second, the approach applies a GIA parametrization in Antarctica utilizing local deglaciation impulse response patterns which are globally consistent. In principle, this GIA parametrization allows to spatially resolve GIA effects in Antarctica, which have not been predicted by GIA forward modelling. Third, the IMC parametrization facilitates a spatial resolution of 50 km which is useful to explore AIS IMC in more detail than results from previous combination studies allow. Fourth, the approach include a parametrization for FAC changes, which allows to circumvent the implementation of a firn density. Finally, the approach enables to incorporate the error covariance information of all input data sets for rigorous accounting of input-data quality limitations. The feasibility of the approach was demonstrated with simulation experiments (Willen et al., 2022).

We present and analyse results from applying this approach over the 10-year observation period from Jan 2011 to Dec 2020 (2011–2021) during which the following data sets are available at the same time: a satellite gravimetry data product from GRACE and GRACE-FO (ITSG-Grace2018 Mayer-Gürr et al., 2018), a satellite altimetry data product from CryoSat-2 (Helm et al., 2014), and changes of FAC derived from RACMO2.3p2 SMB (Wessem et al., 2018) and the IMAU-FDMv1.2A (Veldhuijsen et al., 2023). We validate the results with independent GNSS data.

## 2 Material and Methods

Our aim is to disentangle the IMC of the AIS, which is superimposed with other signals in global gravitational field changes observed by GRACE and GRACE-FO (Chen et al., 2022). From the perspective of Antarctica, gravitational field changes are caused by the global mass redistribution due to AIS IMC, GIA, and far-field

effects from other mass redistributions in the Earth System. We refer the reader to recent review articles (e.g., Hanna et al., 2020; Lenaerts et al., 2019; Whitehouse et al., 2019) for comprehensive background information on the physical processes related to mass changes of the AIS.

## 2.1 Global inversion framework

5 We apply an updated global inversion approach from Willen et al. (2022), which builds upon the work from Rietbroek et al. (2016). The Supplementary Material (SM) provides the updates we have made to the methodology described in Willen et al. (2022) in Sect. A. In the following, we describe the applied inversion methodology. For a more extensive theoretical background and further details about the inversion setup we refer the reader to Willen et al. (2022) and Willen (2023).

10 The global fingerprint inversion from Rietbroek et al. (2016) enables one to partition observed sea level, and to quantify the individual sea level budget components. For this purpose, globally consistent spatial patterns of the individual budget components are derived from a priori information. These spatial patterns serve as fingerprints in the inversion. Scaling factors for the individual fingerprints are then computed via a parameter estimation, utilizing observations from satellite altimetry over the ocean and satellite gravimetry. The quality 15 of the a priori information crucially affects the final result. Rietbroek et al. (2016) found that the scaling factor of the Anatarctic GIA fingerprint in particular was estimated too low, meaning that the GIA effect determined in Antarctica is likely unrealistic.

The inversion approach presented here is designed to resolve this aspect by co-estimating globally consistent AIS IMC and GIA from gravitational-field changes. However, AIS IMC and GIA are superimposed in satellite 20 gravimetry observations, i.e. a spatially resolved parametrization of these signals is strongly correlated and a signal separation appears challenging. For this reason, we additionally introduce satellite observations from ice-sheet altimetry over the AIS, which are sensitive to these signals as well. Furthermore, we make use of products from regional climate and firn modelling to account for ice-sheet surface processes. Thus, this work is an advancement of the work from e.g. Riva et al. (2009), Gunter et al. (2014), Sasgen et al. (2017), and Engels et al. 25 (2018) into a global framework. A globally consistent approach, however, requires a parametrization of far-field effects and cannot treat them by applying regional constraints (Willen et al., 2020). Relevant far-field effects, from the perspective of Antarctica, are the following global mass redistributions: (northern-hemisphere) GIA (Caron and Ivins, 2020), IMC of the Greenland Ice Sheet, glacier mass changes, and terrestrial hydrological mass changes.

30 We formulate the following observation equation including three observational groups, $d$, and and six parameter types, $\beta$:

$$\begin{pmatrix} \boldsymbol{d}^{\text{GRAV}} \\ \boldsymbol{d}^{\text{AIS-ALT}} \\ \boldsymbol{d}^{\text{AIS-FAC}} \end{pmatrix} + \boldsymbol{e} = \begin{pmatrix} \boldsymbol{X}_{\text{GIA}}^{\text{GRAV}} & \boldsymbol{X}_{\text{AIS-IMC}}^{\text{GRAV}} & \boldsymbol{0} & \boldsymbol{X}_{\text{GIS-IMC}}^{\text{GRAV}} & \boldsymbol{X}_{\text{GLAC}}^{\text{GRAV}} & \boldsymbol{X}_{\text{HYD}}^{\text{GRAV}} \\ \boldsymbol{X}_{\text{GIA}}^{\text{AIS-ALT}} & \boldsymbol{X}_{\text{AIS-IMC}}^{\text{AIS-ALT}} & \boldsymbol{X}_{\text{AIS-FAC}}^{\text{AIS-ALT}} & \boldsymbol{0} & \boldsymbol{0} & \boldsymbol{0} \\ \boldsymbol{0} & \boldsymbol{0} & \boldsymbol{X}_{\text{AIS-FAC}}^{\text{AIS-FAC}} & \boldsymbol{0} & \boldsymbol{0} & \boldsymbol{0} \end{pmatrix} \begin{pmatrix} \boldsymbol{\beta}_{\text{GIA}} \\ \boldsymbol{\beta}_{\text{AIS-IMC}} \\ \boldsymbol{\beta}_{\text{AIS-FAC}} \\ \boldsymbol{\beta}_{\text{GIS-IMC}} \\ \boldsymbol{\beta}_{\text{GLAC}} \\ \boldsymbol{\beta}_{\text{HYD}} \end{pmatrix}. \quad (1)$$

The design block matrices, $\boldsymbol{X}$, include the parametrization, i.e. they link the observational groups and parameter types indicated by subscripts and superscripts in Eq. (1).

The three data sets, $\boldsymbol{d}$, are: (1) spherical harmonic coefficients of surface density changes from gravimetry, $\boldsymbol{d}^{\text{GRAV}}$, (2) a grid with surface elevation changes of the AIS, $\boldsymbol{d}^{\text{AIS-ALT}}$, and (3) a grid with changes of the FAC of the AIS, $\boldsymbol{d}^{\text{AIS-FAC}}$. The grid definition is chosen according to the grounded part of the AIS and peripheral glaciers from Mouginot et al. (2017). The six parameter types are: (1) GIA, $\boldsymbol{\beta}_{\text{GIA}}$, (2) ice mass changes of the Antarctic Ice Sheet, $\boldsymbol{\beta}_{\text{AIS-IMC}}$, (3) changes of the firn air content of the Antarctic Ice Sheet, $\boldsymbol{\beta}_{\text{AIS-FAC}}$, (4) ice mass changes of the Greenland Ice Sheet, $\boldsymbol{\beta}_{\text{GIS-IMC}}$, (5) ice mass changes of glaciers outside Antarctica and Greenland, $\boldsymbol{\beta}_{\text{GLAC}}$, and (6) mass changes of non-glaciated water on the continent (e.g., groundwater, surface water), here referred to as hydrological mass changes, $\boldsymbol{\beta}_{\text{HYD}}$.

Here, our intentional goal is the incorporation of error-covariance information, which is a more rigorous approach to address the observational errors than minimizing error effects in the datasets by filtering. Furthermore, the large-scale fingerprints are not sensitive to small-scale errors, such as the typical GRACE/GRACE-FO stripe patterns. The error-covariance matrix, $C(\boldsymbol{d})$, of the observations is

$$C(\boldsymbol{d}) = \begin{pmatrix} C(\boldsymbol{d}^{\text{GRAV}}) & \boldsymbol{0} & \boldsymbol{0} \\ \boldsymbol{0} & C(\boldsymbol{d}^{\text{AIS-ALT}}) & \boldsymbol{0} \\ \boldsymbol{0} & \boldsymbol{0} & C(\boldsymbol{d}^{\text{AIS-FAC}}) \end{pmatrix}. \quad (2)$$

We introduce a GIA-parameterization in $\boldsymbol{X}_{\text{GIA}}^{\text{GRAV}}$ and $\boldsymbol{X}_{\text{GIA}}^{\text{AIS-ALT}}$ to account for GIA-effects in Antarctica and for GIA effects outside Antarctica. In Antarctica, we implement global-consistent GIA fingerprints calculated in response to local deglaciation impulses. These impulse response patterns are generated using the GIA modelling software SELEN[4] (Spada and Melini, 2019) and enable to capture GIA effects independent from GIA forward models with an effective spatial resolution of ∼450 km (Willen et al., 2022). This parametrization does not imply a particular spatial ice-loading history that would predetermine the spatial occurrence of present-day GIA effects.

An exception from forward-model independent GIA parametrization is made on the Antarctic Peninsula.

From our validation experiments, we found that we were not able to retrieve reasonable GIA results for the northern part of the Antarctic Peninsula (Graham Land). We attribute this mainly to the insufficient quality of surface elevation changes derived from radar altimetry here (e.g. Schröder et al., 2019). In turn the significant misfit between GRACE/GRACE-FO products and CryoSat-2 products is captured by an unphysical GIA signal.

This is also the case for other inverse GIA estimates (e.g. Gunter et al., 2014; Engels et al., 2018; Willen et al., 2020). To prevent an unphysical GIA, we decided not to co-estimate GIA in this particular region. We did not include local GIA patterns on the Peninsula in our local GIA-pattern parametrization. Instead we approach the GIA effect here by a global GIA model result which is then subtracted from the observations. We model the GIA effect with an ICE-6G ice history that solely exists in the Graham Land Region by using SELEN[4] (Spada

and Melini, 2019). Figure S1 illustrates the modified GIA parametrization with the Antarctic Peninsula GIA pattern. Admittedly, this GIA pattern has strong limitations to represent the true GIA effect in this region. The upper-mantle viscosity is found to be low here (Nield et al., 2014; Samrat et al., 2021; Ivins et al., 2021). We therefore expect that GIA response time scales are similar to those in the Amundsen Sea Embayment region. This means that the applied pattern (Fig. S1) will only allow an incomplete representation of the actual GIA

and will not resolve GIA effects induced by load changes over the last centuries. Nevertheless, we argue that this methodological adjustment allows to, at least, limit the bias to the entire Antarctic GIA estimate.

Outside Antarctica ("far-field"), we use four global-consistent GIA fingerprints generated with regionally tailored ice loading histories for Greenland, Laurentia, Fennoskandia, and other regions (Patagonia, Barents and Kara Sea, etc.) similar to Rietbroek et al. (2016). Each fingerprint is modelled with SELEN[4] (Spada and

Melini, 2019) using a ICE-6G ice history and VM5a rheology. Additionally, we include two GIA fingerprints to capture a potential residual rotational feedback signal given by GIA-modelling limitations attributed to an erroneous lower mantle viscosity (Caron et al., 2018; Willen et al., 2022).

The parametrization of Antarctic ice mass changes in case of altimetry observations, $X_{\text{AIS-IMC}}^{\text{AIS-ALT}}$, is realized by linking the IMC in a grid cell with its corresponding surface elevation change. This involves, on the one

hand, the elevation change associated with a volume change of pure ice (assuming a density of $917\,\text{kg}\,\text{m}^3$) and, on the other hand, the elevation change caused by the elastic deformation due to the mass change. The latter is obtained while generating the IMC parameterization of the gravimetry observations. $X_{\text{AIS-IMC}}^{\text{GRAV}}$ is created by assuming a point mass change in the centre of each grid cell. We represent this point mass change by a set of spherical harmonic coefficients (Pollack, 1973). The sea-level response of each (ice) point mass change is

calculated by solving the sea-level equation (Farrell and Clark, 1976; Blewitt and Clarke, 2003; Clarke et al., 2005). The point mass change together with its sea-level response is the globally-consistent fingerprint of the ice mass change in each grid cell. The elastic deformation effect of this point mass change is co-calculated when solving the sea-level equation and then implemented in $X_{\text{AIS-IMC}}^{\text{AIS-ALT}}$.

$X_{\text{AIS-FAC}}^{\text{AIS-ALT}}$ and $X_{\text{AIS-FAC}}^{\text{AIS-FAC}}$ are identity matrices, since the grid definitions of the observations and of the parame-

ters are identical. The change in FAC is mapped one-to-one in the altimetry observations. Note that the surface

elevation changes observed by satellite altimetry are parametrized in two parts: the first part is the elevation change associated with ice density and the related elastic deformation. The second part is the change in FAC.

The gravitational-field changes include, as mentioned above, far-field effects (far-field from the perspective of Antarctic mass changes). To take these into account, we introduce GIA parameters to account for GIA effects from outside Antarctica as mentioned above. Furthermore, we introduce parameters for glacier mass changes, $\boldsymbol{\beta}_{\text{GLAC}}$, and continental hydrology mass changes, $\boldsymbol{\beta}_{\text{HYD}}$. The matrices $\boldsymbol{X}_{\text{GIS-IMC}}^{\text{GRAV}}$, $\boldsymbol{X}_{\text{GLAC}}^{\text{GRAV}}$, and $\boldsymbol{X}_{\text{HYD}}^{\text{GRAV}}$ link observed surface-density changes with IMC of the GIS, glaciers, and hydrology, respectively. Glacier mass changes and continental hydrology mass changes are parametrized with 68 and 60 globally consistent fingerprints, respectively (updated according to Uebbing et al., 2019). We extend the hydrology parametriza-tion with a fingerprint to capture the residual hydrological mass change signal over the continents which is not resolved by the applied hydrology parametritzation (cf. Sect. A in SM). The parametrization of IMC in Green-land applies 16 fingerprints for the 8 drainage basins of the Greenland Ice Sheet (Zwally et al., 2012). For this purpose, each basin is divided into a part below and above 2000 m surface elevation, which leads to a total of 16 sub-basins. The mass change of each sub-basin is not assumed to be uniform. Instead, a more realistic mass change pattern within each sub-basin is chosen based on mean rates of surface elevation changes derived from CryoSat-2 satellite altimetry (updated according to Helm et al., 2014). The globally consistent mass change pattern (fingerprint) of each sub-basin is calculated by solving the sea-level equation.

The parameters are estimated by generalized least squares adjustment (e.g., Koch, 1999). We apply a variance component estimation to further optimize the estimates by a relative weighting of the uncertainty information of the three observational groups gravimetry, gravimetry, and FAC changes (cf. Sect. B in SM). Additionally, we implement a Tikhonov regularization of the Antarctic GIA Parameters to prevent unphysical GIA results due to limitations of the error covariances of the input data sets (cf. Sect. C in SM). We solve the following regularized normal equations:

$$\hat{\boldsymbol{\beta}} = (\boldsymbol{N} + \boldsymbol{\Psi}^{\text{T}}\boldsymbol{\Psi})^{-1}\boldsymbol{n} \qquad \text{and} \qquad C(\hat{\boldsymbol{\beta}}) = \sigma^2(\boldsymbol{N} + \boldsymbol{\Psi}^{\text{T}}\boldsymbol{\Psi})^{-1}, \tag{3}$$

where $\hat{\boldsymbol{\beta}}$ are the estimated parameters and $C(\hat{\boldsymbol{\beta}})$ the corresponding covariance matrix. $\boldsymbol{N}$ is the normal equation matrix, $\boldsymbol{n}$ is the right-hand side, and $\boldsymbol{\Psi}$ is the regularization matrix (Tikhonov et al., 1995). We determine the degree of regularization using the L-curve criterion (Hansen, 2001). Sections A–C of the SM include further details about the implemented parameter estimation strategy and closed-loop simulation results to justify the regularization. Furthermore, Section C.2 in the SM provides information on how we choose the *preferred inversion solution* presented in Sect. 3.

## 2.2 Data sets

The observations, $\boldsymbol{d}$, are mean rates according to the time period from Jan 2011 until Dec 2020 (10 years). We use the gravitational field changes ITSG-Grace2018 (Mayer-Gürr et al., 2018; Kvas et al., 2019), which are GRACE/GRACE-FO level-2 products provided as monthly sets of spherical harmonic coefficients up to degree 96 without any GIA correction. We express the gravitational field changes as mass changes in a spherical layer, termed surface-density changes (Wahr et al., 1998). These level-2 products, have a low noise level compared to other products and at the same time almost completely retains the signal (Ditmar, 2022). The gravitational fields are complemented with degree-1 products derived according to Sun et al. (2016). $c_{20}$ coefficients, and $c_{30}$ coefficients in case of GRACE-FO and GRACE accelerometer failures, are replaced with Satellite Laser Ranging products (Loomis et al., 2020). We do not apply any filter to the gravitational fields. It should be noted that GRACE/GRACE-FO level 3 products, e.g. mascon solutions, are not suitable for the investigation presented here due to the following reasons: (1) Mascon solutions are already corrected for the GIA effect, i.e. this GIA correction would have to be back-processed. (2) The globally consistent parametrization cannot applied to level 3 data and would have to be completely re-developed and (3) the rigorous propagation of covariance information would not be possible unless it is available along with the level 3 products.

The surface elevation changes are derived from updated CryoSat-2 products according to Helm et al. (2014). Finally, the FAC changes are derived from the RACMO2.3p2 SMB product (Wessem et al., 2018) and the IMAU-FDMv1.2A firn-thickness change product (Veldhuijsen et al., 2023). Surface elevation and FAC changes are resampled to a grid of $50\,\mathrm{km} \times 50\,\mathrm{km}$ by averaging over a grid cell.

The uncertainty characterization follows Willen et al. (2022). $C(\boldsymbol{d}^{\mathrm{GRAV}})$ is derived from normal equations provided along with the ITSG-Grace2018 gravitational field products. The degree-1 uncertainty is characterized analogous to Willen et al. (2022) based on a degree-1 ensemble over the period under investigation. In the case of $C(\boldsymbol{d}^{\mathrm{AIS\text{-}ALT}})$ and $C(\boldsymbol{d}^{\mathrm{AIS\text{-}FAC}})$, we base the uncertainty characterization on ensembles of surface height rates and FAC rates. The ensemble of altimetry-derived surface elevation changes is based on surface elevation rates derived with altimetry processing techniques, commonly denoted as retrackers, 'EWIDTH', 'ICE1', 'OCOG', and 'TFMRA'. We do not include retrackers in the ensemble, that are known to lead to elevation changes with poor quality. The uncertainty characterization of FAC is based on mean-rate differences of cumulated surface mass balance anomalies derived from the regional climate models RACMO2.3p2 (Wessem et al., 2018) and MARv3.11 (Kittel et al., 2021). Mottram et al. (2021) showed striking differences between products of these two climate models indicating systematic errors. To avoid artificial downweighting of the FAC information within the inversion by too conservative uncertainties, we limit the difference between MARv3 and RACMO2. We specified that the magnitude of the mean-rate difference between RACMO2 and MARv3 of one pixel should not be greater than the mean-rate magnitude derived from RACMO2 for this pixel. Thus, we set a threshold of $100\,\%$ for the maximum deviation in the ensemble of mean rate differences.

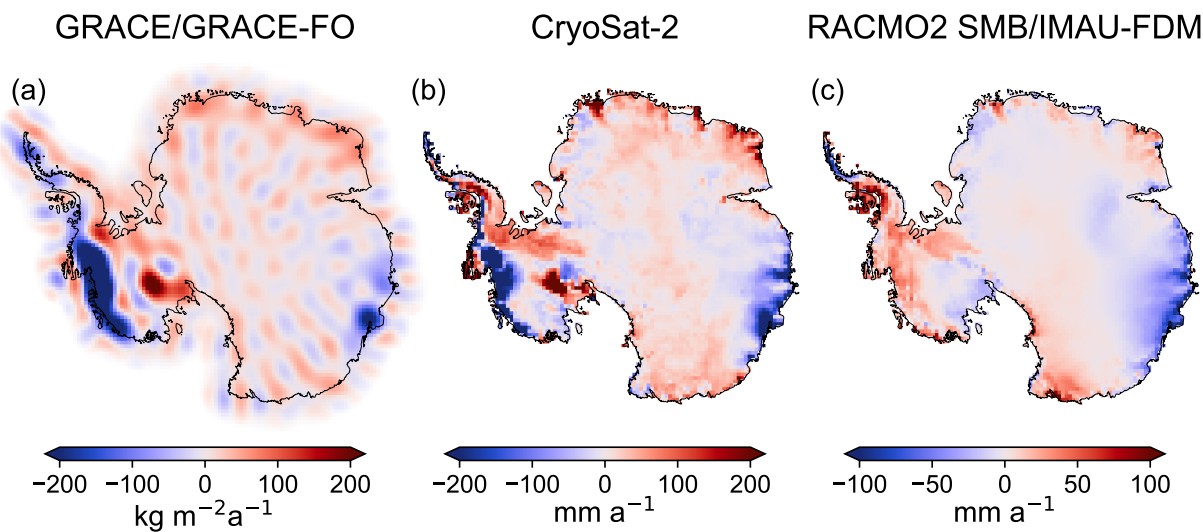

**Figure 1:** Mean rates of data sets for the time period Jan 2011–Dec 2020. (a) GRACE/GRACE-FO-derived surface density rate using ITSG-Grace2018 monthly gravitational fields (Mayer-Gürr et al., 2018) (no filter applied). (b) CryoSat-2-derived surface elevation rate updated according to Helm et al. (2014). (c) Thickness change of firn air content (FAC) derived from RACMO2.3p2 SMB (Wessem et al., 2018) and IMAU-FDMv1.2A (Veldhuijsen et al., 2023).

## 2.3 Assessment methods

We use three approaches to assess the quality and soundness of the results obtained with the described methodology: (1) We compare the resulting GIA estimates with independent data derived from GNSS observations. (2) We compare the Antarctic GIA result with results from an alternative inverse approach as well as from forward modelling. (3) We perform sensitivity tests with alternative input data sets.

In Assessment (1), we use preliminary results from a consistent analysis of more than 270 GNSS stations distributed over entire Antarctica. This analysis has been accomplished in the frame of the SCAR-endorsed Geodynamics In ANTarctica based on REprocessing GNSS dAta INitiative (GIANT-REGAIN, Buchta et al., 2022). Fig. S8 illustrates the locations of the GNSS sites. GIA-related bedrock motion rates from the GNSS data are compared with bedrock motion rates from the inversion results. The bedrock motion time series from the GNSS data are corrected for elastic deformation effects of the solid Earth due to IMC during the observation period of each GNSS site. We determine the IMC using the surface elevation change time series derived from satellite altimetry (Nilsson et al., 2022) and the firn model IMAU-FDM (Veldhuijsen et al., 2023). Based on these IMC, we calculate the elastic-related (vertical) bedrock motion by using the Green's function approach in the spatial domain (Farrell, 1972). We use load Love numbers derived from the Preliminary Reference Earth Model (PREM, Dziewonski and Anderson, 1981). The elastic bedrock motion effect, which is part of the observed surface elevation changes (Nilsson et al., 2022), is accounted for by assuming it to be $-1.5\%$ of the altimetry-derived surface elevation change (Riva et al., 2009). It has to be emphasized that the comparison of

our GIA estimates with those inferred from GNSS serves only the purpose of an independent assessment, and that this data set is not part of the inversion framework. We calculate a weighted root mean square difference (WRMSD) between the GNSS-derived rates and those from the inversion results (INV), as done by Gunter et al. (2014):

$$\text{WRMSD} = \sqrt{\frac{\sum w_i \left( \dot{h}_{i,\text{INV}}^{\text{GIA}} - \dot{h}_{i,\text{GNSS}}^{\text{GIA}} \right)^2}{\sum w_i}}, \tag{4}$$

with the weight, $w$, for each GNSS site, $i$:

$$w_i = \frac{1}{\sigma_{i,\text{INV}}^2 + \sigma_{i,\text{GNSS}}^2}. \tag{5}$$

$\sigma$ indicates the uncertainty (standard deviation) of the estimated rate. We derive $\sigma_{i,\text{INV}}^2$ from $C(\hat{\boldsymbol{\beta}})$ (Eq. 3). We obtain $\sigma_{i,\text{GNSS}}^2$ from the GNSS processing and we additionally assume 10 % of the estimated elastic deformation as its uncertainty.

In Assessment (2), we compare the Antarctic GIA estimate to the modelling result from Caron et al. (2018) and to the regional inverse estimate from Engels et al. (2018). Caron et al. (2018) modelled the present-day GIA effect based on assumptions on the ice loading history and the solid Earth's rheology. They applied a Bayesian inversion approach to find a best-fit GIA model with GNSS observations as well as relative sea level records. This framework includes GNSS observations also in Antarctica from Blewitt et al. (2016). Engels et al. (2018) used a similar data-driven approach, as presented here, to isolate the GIA effect from satellite gravimetry and satellite altimetry. However, the approach is a regional approach, as it regionally constrains GIA by calibrating interim results over a low-precipitation zone. Furthermore, it differs from the result presented here as it utilizes input data from GRACE and ICESat over the time period Feb 2003 to Oct 2009. Engels et al. (2018) incorporated GNSS observations within the estimation procedure to justify parametrization choices, thus making this approach not fully independent from GNSS observations.

For Assessment (3), we run sensitivity tests by using alternative input data sets for the inversion. We use surface elevation changes from Nilsson et al. (2022) as an alternative altimetry product. During the investigated time period, the input data to this alternative product is predominantly from CryoSat-2. The ICESat-2 observations included start only in Oct 2018. Envisat observations are not included for periods later than Sep 2010, that is, not included in the period considered here. Although dominated by the CryoSat-2 input, the alternative surface elevation change estimates by Nilsson et al. (2022) result from a different processing scheme and include data from one alternative altimetry mission to some extent. The uncertainty information provided along with the surface-elevation time series from Nilsson et al. (2022) is less comprehensive than our error characterization for CryoSat-2 products. Since the dataset is largely based on CryoSat-2 data dur-

ing our investigation period, we consider it reasonable to assume the same uncertainty information we use for the CryoSat-2-only data set (Sect. 2.2). Moreover, we test the sensitivity of the results towards a FAC variant by exchanging RACMO2.3p2 SMB with MARv3.11 SMB. Since our ensemble for error characterization was created from differences between MARv3 and RACMO2 SMB, we assume the same uncertainty information,
we apply for FAC changes based on RACMO2.3p2 SMB. In case of gravitational field products, we use solutions from the Center of Space Research of the University of Texas at Austin, CSR RL06.1 products (Pie et al., 2021). For CSR RL06, the predecessor of CSR RL06.1, Ditmar (2022) found the lowest noise level among the Science Data System (SDS) solutions and fair signal retainment. Nevertheless the noise level of CSR RL06.1 is higher than in ITSG-Grace2018 gravitational fields (Fig. S9d). Full error covariance information, likewise to
ITSG-Grace2018, is not provided as a standard product along with CSR RL06.1 products. Pragmatically, we assume ITSG-Grace2018-uncertainty information for CSR RL06.1 gravitational fields here by accepting that the sensitivity test is inconsistent to some extent. Figure S9 provides maps of the alternative input data sets and differences to the input data sets described in Sect. 2.2.

# 3   Results

Figure 2 illustrates the following results from the preferred inversion solution where we apply a regularization parameter of $\varepsilon = 0.3$: the Antarctic GIA-related bedrock motion (Fig. 2a), IMC of the AIS expressed in terms of surface density change (Fig. 2b), and the estimated thickness change of FAC (Fig. 2c). Figure 2d–f shows the associated 2-$\sigma$-uncertainties derived from $C(\hat{\boldsymbol{\beta}})$ (Eq. 3). We intentionally show in Fig. 2a the present-day GIA effect in terms of surface density changes, rather than smoother geoid-height changes as shown elsewhere
(Jiang et al., 2021; Sasgen et al., 2017), in order to demonstrate the limits of the spatial resolution of present-day GIA effects with the inverse approach applied here.

The integrated values are: $86 \pm 21\,\mathrm{Gt\,a^{-1}}$, $-144 \pm 27\,\mathrm{Gt\,a^{-1}}$, and $13 \pm 18\,\mathrm{km^3 a^{-1}}$ in case of AIS GIA, AIS IMC, and AIS FAC, respectively (Table 1). We quantify the apparent mass effect of GIA in order to demonstrate the effect that GIA-induced gravitational field changes would have on gravimetry-only ice mass balance
estimates. This quantification depends on the adopted method to infer ice mass changes from gravitational field changes. Here we convert GIA-related gravitational field changes to equivalent surface mass density changes, as we do it with gravimetry observations (Sect. 2.2, Wahr et al., 1998, Vishwakarma et al., 2022). Subsequently, we integrate over the ice sheet region extended by a 400 km buffer zone. Note that different methodologies of gravimetric IMC inferences would imply different ways of integrating the GIA-equivalent surface mass density
change (Döhne et al., 2023) and that our adopted scheme is not used for our actual IMC estimates but exclusively for expressing our estimated GIA signals in terms of an integrated mass effect. Figure S7 illustrates the Antarctic integrals for all regularization parameters. The FAC change integrated over the AIS is $-4\,\mathrm{km^3 a^{-1}}$ and $13\,\mathrm{km^3 a^{-1}}$ for the input data and the estimate of the preferred inversion solution, respectively. Thus, the

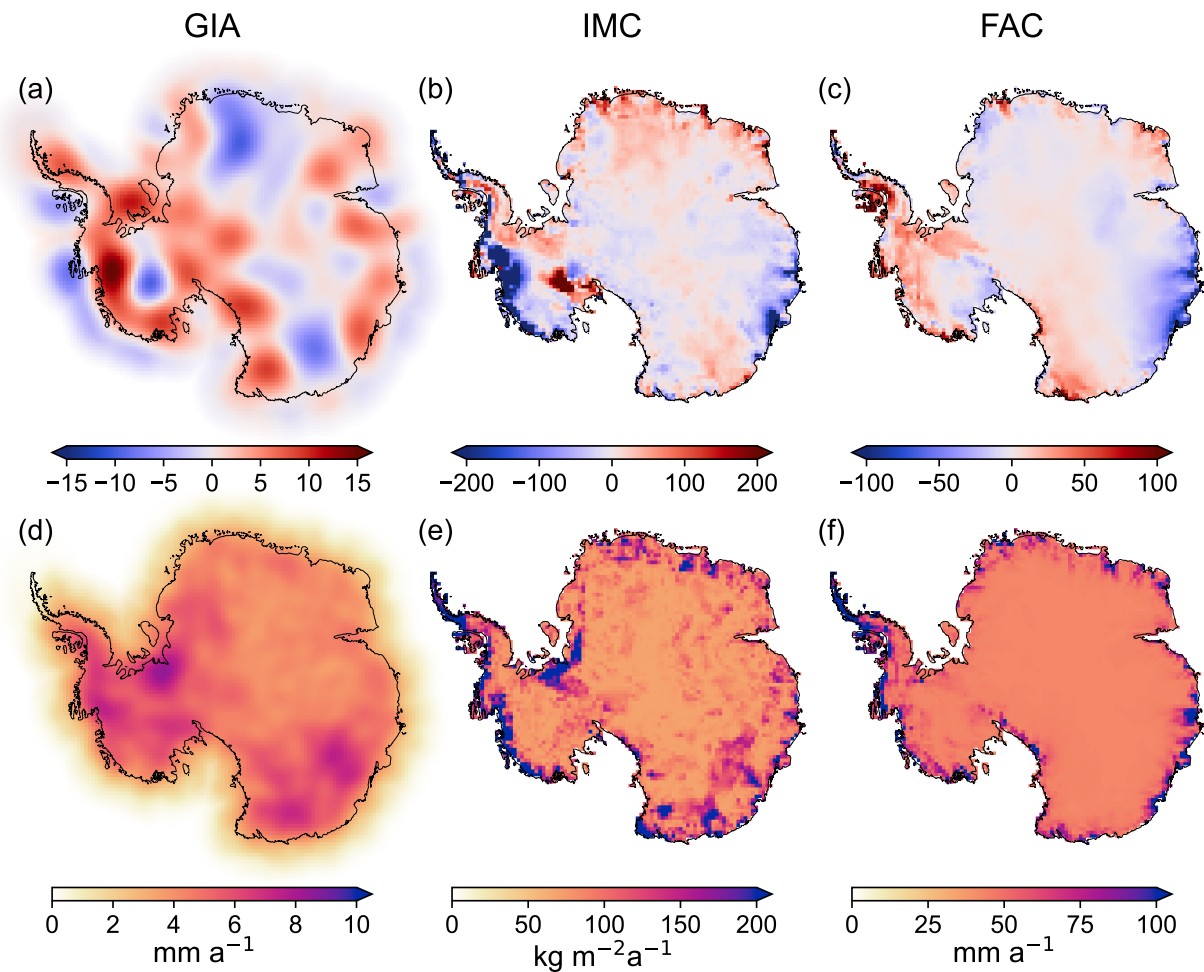

**Figure 2:** Maps of Antarctica illustrating the estimates of a) vertical bedrock motion due to glacial isostatic adjustment (GIA), b) surface density change due ice mass change (IMC) of the Antarctic Ice Sheet (AIS), and c) the thickness change of the firn air content (FAC) derived from the preferred inversion solution. (d–f) shows the 2-$\sigma$-uncertainties, respectively. Units indicated for (d–f) apply columnwise to (a–c).

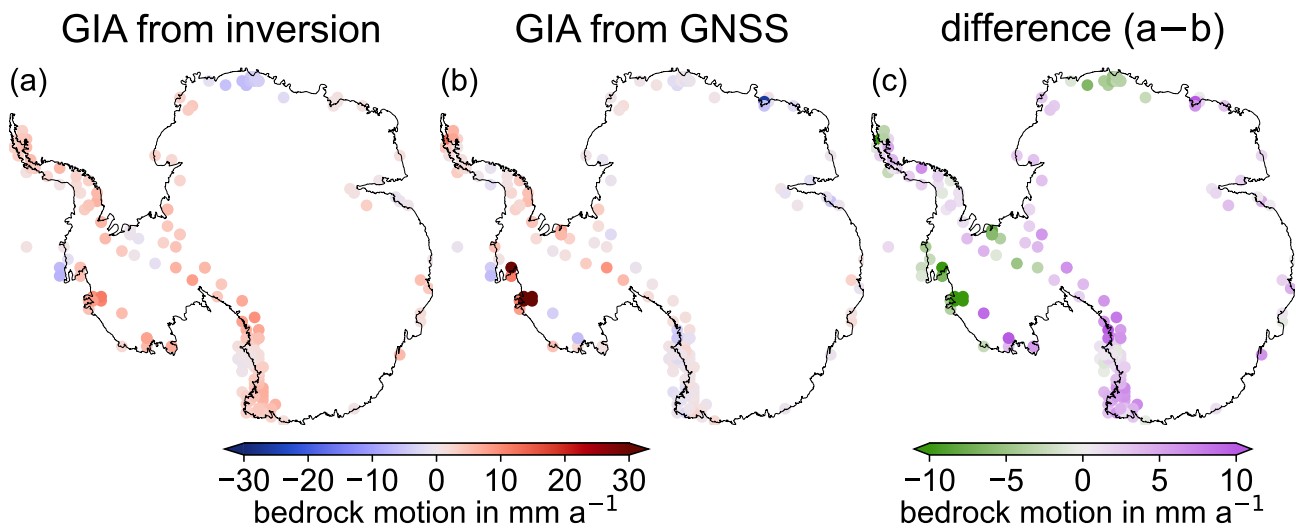

**Figure 3:** Assessment (1): Comparison of GIA-related bedrock motion at GNSS sites from the preferred inversion solution (a), from GNSS observations (b), and the difference between both (c).

input value is still within 2-$\sigma$ uncertainty interval of the estimate ($\pm\,18\,\mathrm{km^3a^{-1}}$). In view of the uncertainty, it is not possible to conclude whether the mean FAC rate for the entire grounded AIS is positive or negative during the 10-year time interval.

In summary we find the following spatial features of IMC: Prominent negative IMC are evident in the
Amundsen Sea Region, Getz Ice Shelf Region, and at Totten and Denman Glacier (Wilkes Land). Positive IMC were detected at Kamb Ice Stream, Ellsworth Land, Dronning Maud Land, Enderby Land, and to some extent at Terre Adélie (cf. Fig. 4a for geographical names). Note that already the input data sets consistently reveal these spatial features for a large part (Fig. 1) and the spatial pattern of the altimetry-derived mean rates (Fig. 1b) basically determines the spatial pattern of the determined AIS IMC (Fig. 2a). The spatial pattern of
FAC change, that enters the inversion as an input data set, is for a large part identical to the pattern of FAC change that is estimated.

For Assessment (1) of these results (Sect. 2.3), Figure 3 provides a spatial comparison of the estimated GIA bedrock motion of the preferred inversion solution and the GNSS-derived bedrock motion rates. The WRMSD (Eq. 4) is 4.9 mm/a. For Assessment (2), Figure 4 illustrates maps of the GIA estimate from the preferred
inversion in comparison with the GIA results from Caron et al. (2018) and Engels et al. (2018).

In agreement between the GIA result of the preferred inversion solution and the alternative GIA inverse estimate from Engels et al. (2018) (Fig. 4b) and the optimized forward modelling result from (Caron et al., 2018) (Fig. 4d) are the bedrock uplift in Ellsworth Land, and somewhat in the Ross-Ice-Shelf region, Filchner-Ronne-Ice-Shelf region, and Wilkes Land (cf. Fig. 4a for geographical names). However, there are differences in the
determined magnitudes (Fig. 4c+e). The comparison reveals some common features of the inverse estimates

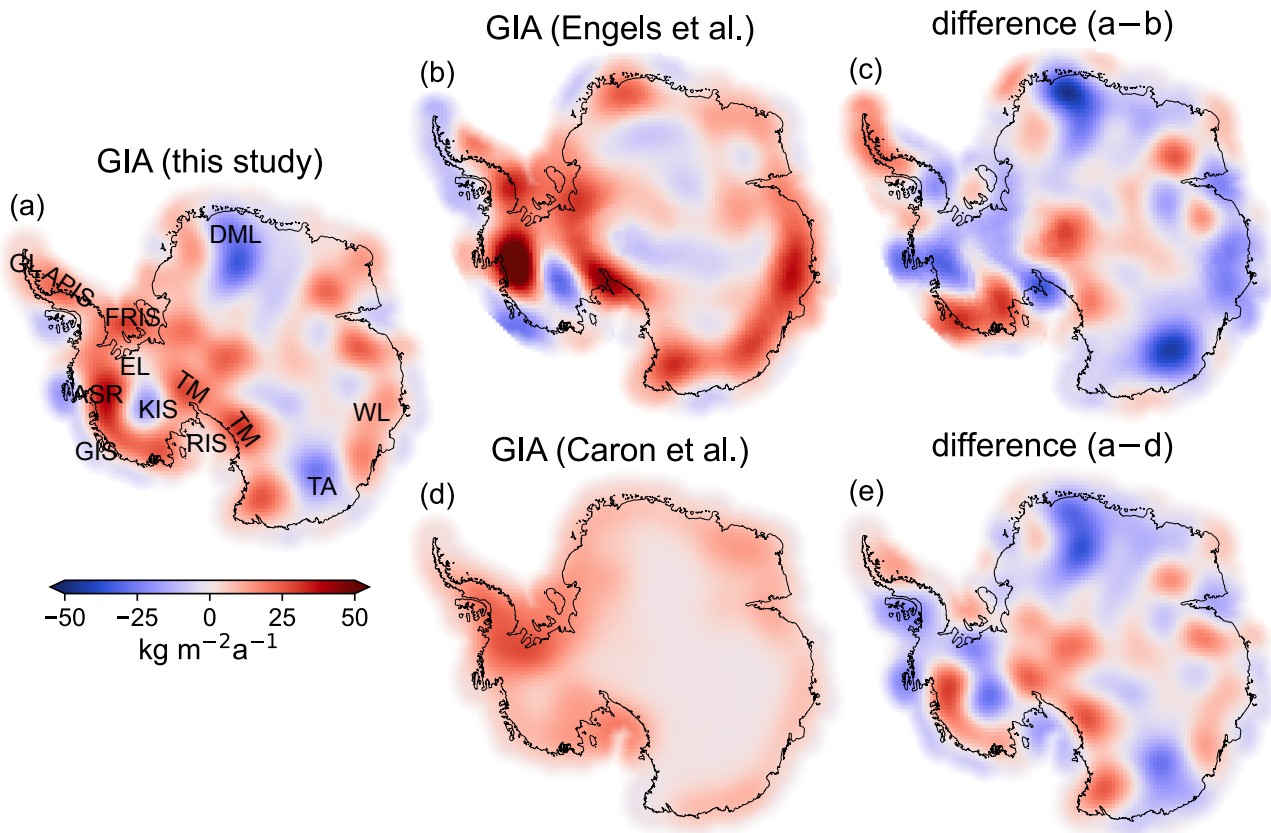

**Figure 4:** Assessment (2): The GIA-related surface density rate from the preferred inversion solution (a), from Engels et al. (2018) (b), and from Caron et al. (2018) (d). The deviation of the latter two from the preferred inversion solution is illustrated in (c) and (e), respectively. The integrated GIA mass effects are $86\,\mathrm{Gt\,a^{-1}}$ (a), $146\,\mathrm{Gt\,a^{-1}}$ (b), and $117\,\mathrm{Gt\,a^{-1}}$ (d) using a $400\,\mathrm{km}$ offshore buffer zone (Gunter et al., 2014). In (a) the following geographical names are labelled: Amundsen Sea Region (ASR), Antarctic Peninsula (APIS), Dronning Maud Land (DML), Ellsworth Land (EL), Filchner Ronne Ice Shelf (FRIS), Getz Ice Shelf (GIS), Graham Land (GL), Kamb Ice Stream (KIS), Ross Ice Shelf (RIS), Terre Adélie (TA), Transantarctic Mountains (TM), and Wilkes Land (WL).

(Fig. 4b, Engels et al., 2018) that are not found in the forward modelling result (Caron et al., 2018) (Fig. 4d): These are the bedrock uplift in the Amundsen Sea region, bedrock uplift in the Transantarctic Mountains, and bedrock subsidence in the Kamb Ice Stream area. However, between the two GIA inverse estimates the local magnitudes and the spatial assignments of features differ (Fig. 4c). The bedrock subsidence indicated by the preferred inversion solution in Dronning Maud Land and Terre Adélie are not part of the GIA results reported by Engels et al. (2018) and Caron et al. (2018). Engels et al. (2018) identified GIA-induced bedrock subsidence at the Antarctic Peninsula and in the Getz Ice Shelf region that are not part of the GIA estimate of the preferred inversion solution and of the GIA forward modelling result (Caron et al., 2018).

WRMSD (Eq. 4) between GNSS observations and the GIA estimate from the preferred inversion solution (Fig. 3), from Engels et al. (2018), and from Caron et al. (2018) are 5.3, 6.8, and 6.6 mm/a, respectively. Note that we transferred the GIA-related surface density change from Engels et al. (2018) and Caron et al. (2018) to bedrock motion using a GIA density mask similar to Gunter et al. (2014). These three WRMSD do not include any weight for GIA-related uncertainties, but only for GNSS-related uncertainties, because there is no consistent GIA uncertainty information for the three GIA estimates available. By not including the GIA uncertainty in the weights, we ensure that all three GIA models are treated equally for comparison with GNSS. For this reason the WRMSD values differ from the WRMSD values given above and the values illustrated in Fig. S5.

For Assessment (3), Figure 5 provide maps of the sensitivity test results in Antarctica. Differences between integrated results from the preferred inversion solution and integrated results from the sensitivity tests are smaller than the estimated 2-$\sigma$-uncertainties. Table 1 summarizes the results of the sensitivity tests. Inte-

**Table 1:** Comparison of integrated GIA mass effect in Antarctica (Antarctic GIA), integrated ice mass change of the Antarctic Ice Sheet (AIS IMC), and firn air content volume change of the Antarctic Ice Sheet (AIS FAC) that result from the sensitivity tests using alternative input data sets. 'preferred solution' are results from the preferred inversion solution based on ITSG-Grace2018 gravitational fields, CryoSat-2-derived surface elevation changes, SMB from RACMO2.3p2, and firn thickness changes from IMAU-FDM. 'alt_JPL' is based on surface elevation changes from Nilsson et al. (2022). 'smb_MAR' uses FAC changes derived from MARv3.11 SMB (Kittel et al., 2021). 'grav_CSR_ITSG_err' utilizes gravitational fields from CSR RL06.1 (Pie et al., 2021) and includes the uncertainty information from ITSG-Grace2018. Values in brackets are the deviation from the preferred solution result (first row). Indicated uncertainties are 2-$\sigma$-values derived from Eq. (3). Figure 5 provides maps of the sensitivity results.

| | Antarctic GIA in $\mathrm{Gt\,a^{-1}}$ | AIS IMC in $\mathrm{Gt\,a^{-1}}$ | AIS FAC in $\mathrm{km^3a^{-1}}$ |
|---|---|---|---|
| preferred solution | $86 \pm 21$ | $-144 \pm 27$ | $13 \pm 18$ |
| alt_JPL | $101 \pm 21$ | $-162 \pm 26$ | $9 \pm 18$ |
| | $(+15)$ | $(-19)$ | $(-4)$ |
| smb_MAR | $85 \pm 22$ | $-142 \pm 28$ | $-1 \pm 25$ |
| | $(-1)$ | $(+1)$ | $(-14)$ |
| grav_CSR_ITSG_err | $61 \pm 32$ | $-113 \pm 45$ | $9 \pm 25$ |
| | $(-25)$ | $(+30)$ | $(-4)$ |

grated values of the Antarctic GIA mass effect, the AIS IMC, and the change in FAC are compared. In each experiment, one of the three data sets has been substituted by an alternative data product.

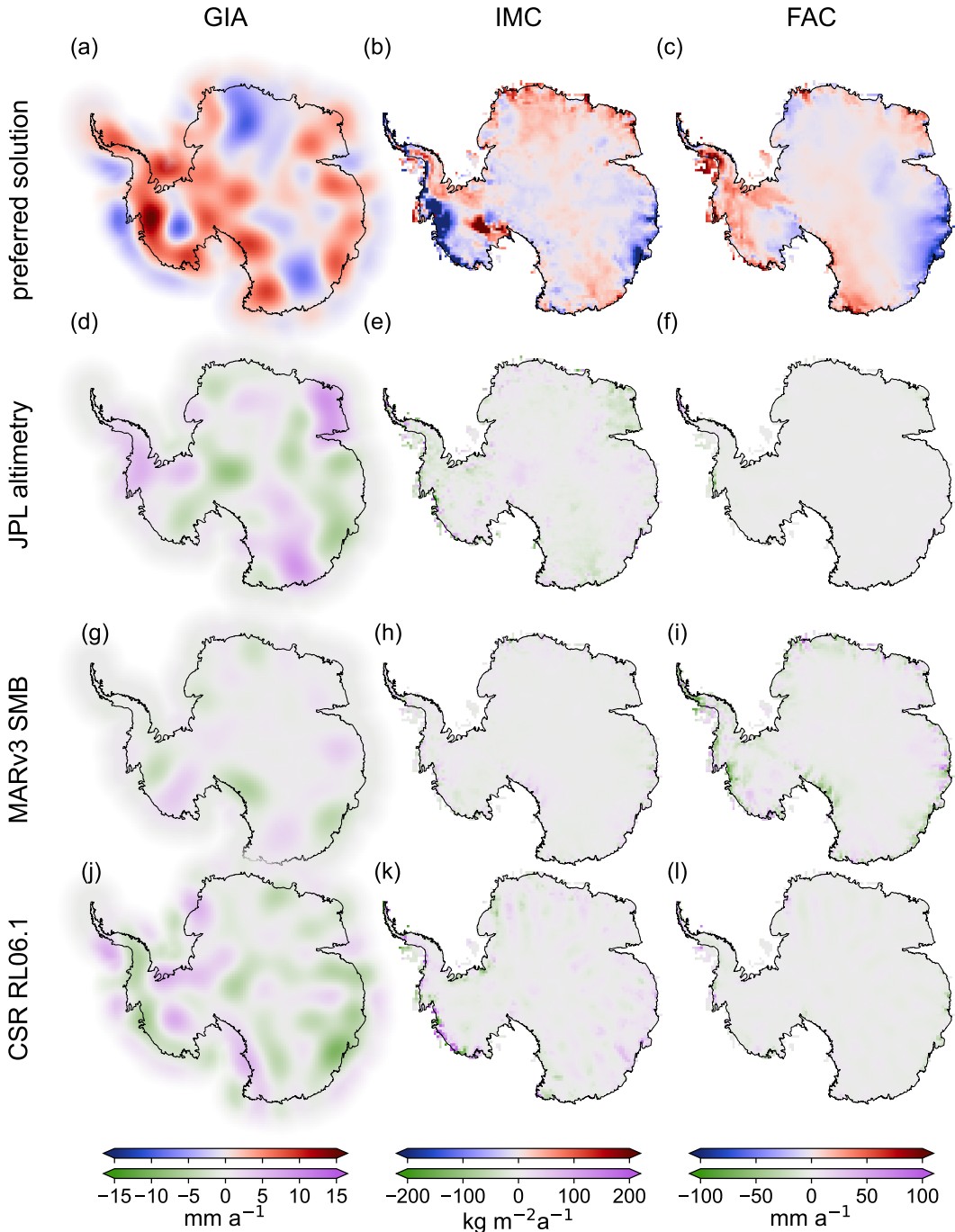

**Figure 5:** Assessment (3): Results from sensitivity tests using alternative input data sets (Fig. S9). The first row (a–c) shows the results from the preferred inversion result (Fig. 3). The second row (d–f) shows the differences between the inversion results and the first row where the surface elevation rate from (Nilsson et al., 2022, Fig. S9b) is used (test result minus preferred inversion solution). Similar, the third row (g–i) and fourth row (j–l) show differences to the inversion results with MARv3.11 SMB data (Kittel et al., 2021, Fig. S9c) and CSR RL06.1 gravitational fields (Pie et al., 2021, Fig. S9a), respectively.

# 4 Discussion

## 4.1 Assessment and comparison

The AIS mass balance of $-144 \pm 27\,\mathrm{Gt\,a^{-1}}$ (2011–2021) from the preferred inversion solution is in the range from $-94$ to $-202\,\mathrm{Gt\,a^{-1}}$ (2010–2019) given in the 6[th] IPCC Assessment Report (Fox-Kemper et al., 2021b). Our estimated AIS contribution to global mean sea level for 2011–2021 is $0.40 \pm 0.07\,\mathrm{mm\,a^{-1}}$. This is very close to the result from the most recent IMBIE study (Otosaka et al., 2023b) at $0.40 \pm 0.09\,\mathrm{mm\,a^{-1}}$ for the same decade. It is worth noting that the Antarctic result from Otosaka et al. (2023b) is based on a total of 23 different ice mass balance estimates. The estimated integrated GIA effect of $86 \pm 21\,\mathrm{Gt\,a^{-1}}$ is at the upper limit of integrated values presented by Whitehouse et al. (2019) and Shepherd et al. (2018), but lower than the results by Caron et al. (2018) and Engels et al. (2018). The stated uncertainties are also plausible. The differences of the sensitivity results to the reference result of the Antarctic-wide integrated values are always smaller than the 2-sigma uncertainties derived from the estimate (Table 1). From this, we conclude that the accuracy of the presented integrated results is sound.

As described in Sect. 3, the comparison of the preferred solution with alternative GIA results reveals some similarities but also prominent differences. The GIA result of the preferred solution fits better to GNSS observations than the GIA solutions by Engels et al. (2018) and Caron et al. (2018) (Fig. 4). However, GNSS-derived bedrock motion is only available for some parts of the Antarctic continent. The comparison is therefore always subject to the asymmetry given by the spatial coverage of GNSS data. In addition, the GIA result of the preferred solution significantly underestimates the bedrock uplift observed with GNSS in the Amundsen Sea region (Fig. 3c), which is presumably more realistically imaged in the result by Engels et al. (2018).

Other data combination approaches, that aim to estimate present-day GIA effects, found significant GIA-induced bedrock subsidence in the Getz Ice Shelf region. Such subsidence is apparent in the GIA estimates according to Sasgen et al. (2017), Engels et al. (2018) (somewhat offshore in Fig. 4b), and Riva et al. (2009) (clipped by the choice of the colourbar limits in Fig. 3a in Riva et al. (2009), but visible in Fig. 2f in Martín-Español et al. (2016)). Before we implemented the IMC and FAC parameterization in the peripheral glacier regions, we also obtained this negative anomaly (Fig. 6.3 in Willen, 2023). This negative GIA anomaly vanishes, by extending the IMC and FAC parametrization to include the peripheral glacier regions.

## 4.2 Methodological implications

In simulation experiments by Willen et al. (2022), GIA could be spatially resolved without filtering or regularization. This requires profound knowledge about error covariances of the input data sets. We find that the used error covariances of the available input data sets (Fig. S12) is limited and it is not useful to determine realistic Antarctic GIA effects (Fig. S6d+g) solely by relying on the error covariance information. In particular,

Figure S12e illustrates that the error characterization of the altimetry trends, based on a data processing ensemble, leads to the result that the errors are strongly correlated on continental scale, i.e. they represent a bias. It is realistic that the error covariance information comprises biases, but whether we capture them sufficiently remains questionable. Note that it is not possible to account for locally or regionally limited errors by including these continental scale error patterns in the parameter estimation.

However, with further simulations (SM) we demonstrated that a regularization can help to somewhat compensate this lack of knowledge and that it is possible to derive physically plausible results, especially in terms of integrated values (Fig. S3). Note that the found regularization optimum from the simulations cannot be applied to define the regularization parameter in the real-data case. This parameter is defined by the L-curve criterion discussed in the next paragraph. The simulation demonstrates, that the noise level of the results is high, especially in case of IMC (Fig. S4k). Applying the regularization on Antarctic GIA parameters and neglecting correlated errors of the input data sets are the main methodological limitations of the work presented here. In order to spatially resolve GIA independent from forward-models, we need to accept for the moment an enhanced sensitivity towards input data errors.

For the results presented here, we can avoid classical filtering of the input data, such as Gaussian smoothing or decorrelation filtering (e.g., Swenson and Wahr, 2006) in case of GRACE/GRACE-FO data, as it has been done in other data combination studies (e.g., Gunter et al., 2014). It is not necessary to equalize the spatial resolution of the datasets prior the joint inversion, i.e. we can avoid coarsening the spatial resolution of the input datasets. As discussed above, we implement a regularization of the Antarctic GIA parameters to prevent dominant spatial oscillations that otherwise appear in the GIA result and cannot be physically justified (Fig. S6). Regularization is, likewise Gaussian filtering is, methodologically less advanced than capturing errors by including the error covariance information. Nevertheless, we can justify the choice of a regularization parameter of $\varepsilon = 0.3$—the amount of damping—with two arguments in the real data case. First, it can be justified by the L-curve criterion (Fig. S5a). Second, the bend in the L-curve coincides with inversion solutions that show the smallest deviation from independent GNSS observations in terms of WRSMD (Eq. 4). Despite the regularization, the GIA result of the preferred inversion solution shows spatial oscillations (Fig. 2a), which are anti-correlated to the IMC result to some degree (Fig. 2b, Fig. S14).

The implemented hydrological residual fingerprint (Fig. S2) allows to capture possible far-field effects due to the limitations imposed by the imperfect hydrology parametrization. Evaluated over the Antarctic continent with a 400 km buffer zone the integrated mass effect of the hydrological residual fingerprint amounts to $-7.4 \, \text{Gt} \, \text{a}^{-1}$. Nevertheless, applying this fingerprint could be only an interim solution that we use here in an Antarctica-focussed study. As soon as an improved globally consistent hydrology parametrization is available, this caveat can be remedied.

### 4.3 Interpretation

#### 4.3.1 GIA estimate in East Antarctica

Limitations in spatially resolving GIA in Antarctica are indicated by the anticorrelation of some patterns of the IMC result and the GIA results (Fig. 2a+b, S13 and S14). For East Antarctica (with its rheology favoring GIA response times of millennia), we do not expect such anticorrelation for the actual signal of IMC and GIA, because such anticorrelation would require an associated correlation between patterns of deglaciation on millennial time scales and present-day IMC. Rather, the resolved GIA and IMC patterns in East Antarctica (Fig. 2a+b) indicate spatial error patterns propagated from the input data (Fig. 1a+b, S11e, 5). The sensitivity experiments (Assessment 3) show that the GIA signal in Terre Adélie and Wilkes Land (Figure 4) depends on the choice of the altimetry product. In Wilkes Land it also depends significantly on the choice of the gravimetry product. This is obviously due to the non-consideration of correlated errors within the parameter estimation. Moreover, the simulation experiments (SM), where we test the regularization of Antarctic GIA, reveal that correlated altimetry errors are obviously reflected in the GIA and IMC result (Fig. S4j+k). This is also evident from the larger RMS error we find for AIS IMC than for the experiment where we have full knowledge on error covariance information. Nevertheless, the integral is very close to the simulated truth. From this we conclude that the preferred inversion solution still contains GIA and IMC patterns, which are artefacts due to data quality limitations rather than resolved physical GIA and IMC signals. This means: One should be cautious when interpreting the short-scale spatial GIA patterns in East Antarctica by physical means. Nevertheless, predictions from GIA forward modelling disagree here, too, because there is a lack of knowledge in ice loading history and the rheological structure (Whitehouse et al., 2019). Given this, it is currently challenging to ascertain how significant the identified spatial patterns of GIA-related bedrock motion in the interior of East Antarctica are in terms of physics. This implies that we cannot decide how useful our East-Antarctic GIA estimate is as a boundary information for testing glacial histories or rheological models. Nevertheless, the integrated values from the GIA estimate and thus the large-scale effects may hold some promise for this task. Measurements of the bedrock motion beneath the East Antarctic Ice Sheet would be helpful as an independent information.

#### 4.3.2 GIA estimate in West Antarctica

The spatial resolution capability of the chosen parametrization is, in a best case, based on reasonable physics. However, if the parametrization is at a finer resolution than the resolution capability of the data allows, this leads to overfitting in the inversion. Willen et al. (2022) demonstrated that the GIA parametrization applied here, which was chosen in agreement with the spatial resolving capability of solely GRACE/GRACE-FO data, is not able to resolve GIA effects associated with low upper-mantle viscosity and ice loading changes over the last centuries. Such GIA effects, as postulated for the Antarctic Peninsula and the Amundsen Sea region

(Nield et al., 2014; Barletta et al., 2018; Samrat et al., 2021), require a spatial resolution capability of $\sim$100 km (gravitational fields up to degree $\sim$200). The GNSS comparison with the GIA result of the preferred inversion solution illustrates the apparently limited GIA-imaging capability within the Amundsen Sea region (Fig. 3). Furthermore, the regularization dampens the GIA signal. In summary, with the inversion approach presented here, we are not able to fully spatially resolve GIA effects associated with low upper mantle viscosity. What we present here is de facto a smoothed version of the true GIA signal. For comparative studies with forward modeling results that aim to represent the realistic rheological structure in West Antarctica, the comparison of smoothed results could at least help to constrain the parameter space. A high-resolution observation-based determination of GIA in this region remains a task for future work. This holds for the Antarctic Peninsula, too, where the GIA result presented here equals a classical GIA modelling result as described in Sect. 2.1.

### 4.3.3 IMC estimate

The spatial patterns of IMC are essentially determined by satellite altimetry (Sect. 2.2), which enables the high spatial resolution based on the selected parametrization. It is noteworthy that this IMC estimate was determined globally consistently and reconciles GRACE/GRACE-FO and CryoSat-2 observations in a least-squares sense. Satellite gravimetry and altimetry are traditionally used separately to determine IMC and have differed significantly in IMBIE assessments (Otosaka et al., 2023b). The result presented here is also in excellent agreement with the estimate from the statistical analysis of 23 different mass balances assessed in IMBIE. This lends confidence to our results and, hence, to the applied method. However, it is noteworthy that, as mentioned in Section 4.3.1, the spatial IMC features are partly anti-correlated with some of the found GIA features which we deem unphysical. This is also reflected in the results of the sensitivity experiments (Fig. 5). This means that not all resolved spatial patterns can be interpreted as IMC. Based on the sensitivity experiments, we conclude that the stated uncertainty of 27 Gt a$^{-1}$ (2-$\sigma$) is realistic. However, this uncertainty is still large considering that it amounts almost $\sim$20 % of the magnitude.

### 4.3.4 FAC estimate in context of its uncertainty

We apply the FAC uncertainty information which assumes that differences between RACMO2.3p2 and MARv3.11 SMB products represent the true modelling error and can be used to characterize the SMB uncertainty. If we apply this empirical uncertainty information, this leads to unphysical GIA artefacts. In addition to ignoring correlations, we constrain the characterization of uncorrelated FAC errors. We presume that the empirical FAC uncertainty information is not fully sufficient to account for the true but unknown FAC error. Especially MARv3.10 SMB shows a striking difference from the ensemble mean SMB in the (leeward of) Transantarctic Mountain region (Figure 6f, (Mottram et al., 2021)), where we found unphysical GIA in preliminary results. As we use differences between RACMO2.3p2 and MARv3.11 SMB products to characterize the FAC uncertainty,

this ends up in a large empirical uncertainty assumption here. In turn, the uncertainty assumption in this region allows unrealistic liberty within the inversion framework to explain the data. In fact, we presume that the spatial pattern of the differences (Figure 6f, (Mottram et al., 2021)) propagate to the GIA estimate presented here. For this reason, we constrain the mean rate ensemble from which we derive the FAC uncertainty as described in Sect. 2.2. Future studies may show the degree of improvement of FAC changes that can be achieved, if a more sophisticated uncertainty characterization of FAC is available (Kappelsberger et al., 2023). Systematic SMB modelling errors, however, only explain part of the unphysical GIA effects of the preliminary results, as these also occur if the error covariance information of the other data sets is incorporated.

## 4.4 Outlook

If there is no improved error covariance information available, the spatial error patterns in the results could also be damped, for example, by applying Gaussian smoothing to the input data (instead of applying a regularization). This may lead to IMC results comparable to usual gravimetric mass balances (Groh and Horwath, 2021), but would, however, smooth the entire result. An alternative strategy may be to adjust the regularization depending on the region, e.g. by implementing a stronger regularization of the GIA effects within East Antarctica where they are presumably small (Whitehouse et al., 2019). This needs to be justified based on additional information. Likewise a GIA parametrization, that is more oriented towards forward modelling results, may be used, but it has been the very intention here to make the estimation independent from possible GIA modelling errors.

The implementation of an extended IMC parametrization could further optimize the inversion result. A parametrization would be desirable that allows for fine spatial resolution only where it is justified by the signal-to-noise ratio (SNR) of the input data. The spatial resolution given by the parametrization may be fine where the SNR is large and coarse where the SNR is small. Further, fine spatial resolution is only needed where mass change processes occur on small spatial scales, i.e. in particular the ice sheet margin. The IMC estimate (Fig 2b) indicates that the inversion is good at spatially separating large IMC amplitudes, e.g. at the ice sheet margin where ice-dynamic flow changes govern IMC. This is probably not necessary in the East Antarctic interior, where small-scale IMC changes are less likely to be relevant. Such an adapted IMC parametrization could help to reduce the presumed error patterns in East Antarctica (Fig. S12e), which are currently erroneously assigned to a GIA effect in the inversion (Sect. 4.3).

We expect a significant quality improvement of satellite altimetry-derived surface elevation changes with new retracking methods in case of radar altimetry (Helm et al., 2023) and with the growing availability over time of laser altimetry products from the ICESat-2 mission. In terms of mean rates, the quality of the results in general will grow by investigating longer time periods. For instance, it is expected that the enhanced data products will facilitate the observation-based assessment of the GIA effect on the Antarctic Peninsula, which

was not achieved in this study.

In a next step, the approach used here could be extended so that IMC changes can be resolved in time and not just as mean rates over defined time periods. According to the input data set availability, monthly IMC and FAC changes may be estimated. This requires to characterize uncertainties on the same temporal scales.

Moreover, the applied methodology is designed to serve as a complement for the global inversion of all sea level contributions (Rietbroek et al., 2016; Uebbing et al., 2019) and may allow to resolve issues while co-estimating the GIA component.

# 5  Conclusions

We demonstrated the successful application of a joint global inversion approach with a focus on Antarctica. It combines the advantages of data sets derived from GRACE/GRACE-FO, CryoSat-2, and regional climate and firn modelling. We claim that the results presented have the following advantages over previous studies: The estimation procedure preserves global consistency in its representation of mass changes, because a global framework can avoid regional constraints as implemented in previous inverse GIA investigations. The inversion enables to spatially resolve GIA effects in Antarctica largely detached from GIA forward modelling constraints. In addition, it enables to determine high-resolution (50 km) IMC. FAC changes are implemented in the parameter estimation procedure instead of taking them into account deterministically only. Lastly, the estimation procedure uses a weighting based on realistic input-data uncertainties and, thus also allowing sound uncertainty estimates of the results.

We estimate the following Antarctic-wide integrated values over the 10-year time interval from Jan 2011 until Dec 2020: a present-day GIA mass change effect of $86 \pm 21\,\mathrm{Gt\,a^{-1}}$; AIS IMC of $-144 \pm 27\,\mathrm{Gt\,a^{-1}}$ and volume change of FAC of $13 \pm 18\,\mathrm{km^3 a^{-1}}$. IMC and FAC integrals include peripheral glaciers. The GIA integral includes a 400 km offshore buffer zone. From the comparison with other published AIS IMC and GIA results, from the comparison with independent GNSS observations, and from sensitivity tests we conclude that the presented results are sound and the provided accuracy is reliable. We found better agreement of the GIA results from the preferred inversion solution with independent GNSS observations than GIA results from other inversion studies that even incorporate GNSS observations in their estimation procedure.

Willen et al. (2022) and this study demonstrated the relevance of having profound knowledge on error covariances of the input data sets available. So far, we were not able to completely eliminate spatial error patterns in the results which propagate from the input data. Moreover, we can attribute error sources only to some extent.

We see potential for improvement of the approach applied here by advancing the global hydrology parametrization, including improved error-covariance information, and further developing the IMC and GIA parametrization.

## Data availability

GRACE/GRACE-FO monthly gravitational fields can be obtained via http://icgem.gfz-potsdam.de/series

CryoSat-2 data: https://earth.esa.int/eogateway/catalog/cryosat-products

RACMO2 SMB and IMAU-FDM on reasonable request (https://www.projects.science.uu.nl/iceclimate/)

5    MARv3.11: https://zenodo.org/record/4459259

## Author Contributions

CRediT: Conceptualization: MW, MH | Data curation: VH, EB, BU, MW, MS | Formal analysis: MW | Funding acquisition: MH, JK, MS | Investigation: MW | Methodology: MW, MH | Software: MW | Supervision: MH | Validation: MW | Visualization: MW | Writing - original draft: MW | Writing - review & editing: MH, EB, 10  VH, BU, JK, MS, MW

## Competing interests

The authors declare that they have no conflict of interest.

## Acknowledgements

We thank the three anonymous referees for their very helpful feedback which improved the manuscript. The 15  work of MW and BU was funded by the grants HO 4232/4-2 and KU 1207/22-2, respectively, "Reconciling ocean mass change and GIA from satellite gravity and altimetry (OMCG)" (project number 313917204) of the Deutsche Forschungsgemeinschaft (DFG) as part of the Special Priority Program (SPP) 1889 "Regional Sea Level Change and Society" (SeaLevel). The work of EB was funded by the grant SCHE 1426/26-1 and 2 (project number 404719077) of the Deutsche Forschungsgemeinschaft (DFG) as part of the SPP 1158 "Antarc- 20  tic Research with Comparative Investigations in Arctic Ice Areas". We kindly thank all colleagues and institutions who provided geodetic GNSS data in Antarctica to the SCAR-endorsed Geodynamics In ANTarctica based on REprocessing GNSS dAta INitiative (GIANT-REGAIN) led by Mirko Scheinert and Matt King (University of Tasmania, Hobart, Australia).

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
