# Peer review of "Globally consistent estimates of high-resolution Antarctic ice mass balance and spatially-resolved glacial isostatic adjustment"

_The Cryosphere, 2023_

## Referee Comment (RC2)

**Review report**

**Manuscript Number:** tc-2023-119.

**Full Title**: Globally consistent estimates of high-resolution Antarctic ice mass balance and spatially-resolved glacial isostatic adjustment

**Authors:** Matthias O. Willen, Martin Horwath, Eric Buchta, Veit Helm, Brend Uebbing, and Jürgen Kusche

Submitted to ***The Cryosphere***

**Recommendation:** Minor revision

In this paper, the authors present a successful joint global inversion approach with a primary focus on Antarctica's mass balance. Utilizing diverse datasets, including GRACE/GRACE-FO, CryoSat-2, regional climate modeling, and firn modeling, the study analyzes the spatial characteristics of the Antarctic GIA component (Glacial Isostatic Adjustment). A distinguishing feature of this work is its deviation from previous studies, which predominantly relied on the GIA forward modeling constraints. The paper further indicates its capacity to discern changes in ice mass balance at a high spatial resolution (50 kilometers). Additionally, the authors delve into the quantitative evaluation of result uncertainty, a pivotal component in such analyses, by incorporating weightings that hinge on the input data's uncertainty.

The introductory section of the paper effectively provides a comprehensive overview of the paper's objectives, theoretical framework, and relevance in the context of prior research. This well-structured introduction serves as a solid foundation, elucidating the research's objectives for the readers.

Analysis of Results and Interpretations in Section 4.3, devoted to analyzing results and their interpretations, warrants improvement in structure and clarity. Reorganizing this section, employing clear and concise language, logical arrangement, and distinct paragraphs for each subject, is recommended. These enhancements will significantly enhance the comprehensibility of the information presented, aiding readers in deriving meaningful insights.

The paper is praiseworthy for its meticulous approach to data handling and analysis, which has led to high-quality findings. The results presented in the form possess substantial scientific value, rendering them well-suited for publication in ***The Cryosphere.***

In conclusion, with appropriate improvements to the logical structure, particularly within section 4.3, and minor revisions, this study can significantly contribute to ***The Cryosphere***. This paper, including the robustness of the results and the comprehensive analysis of various datasets, is suitable for publication.

Below, we will comment on section 4.3 and list technical minor points on this manuscript.

**About section 4.3:**

In Section 4.3, the discussion primarily centers on the correlation between the spatial patterns derived through the current approach and the interplay between the analysis methodology and its outcomes. Furthermore, this section

aims to enhance the physical comprehension of each dataset about Antarctica in this thesis. Critical objectives for this section include characterizing the spatial distribution of Antarctic Glacial Isostatic Adjustment (GIA) as determined by the current approach, summarizing the distinctive attributes and benefits of the present methodology, and offering physical interpretations. This entails a comparative analysis with forward modeling and other analytical techniques that have historically been utilized for separating Antarctic GIA and interpreting the physical processes of each result related to Antarctic mass balance.

Moreover, this section should touch upon the potential for constraining uncertainties within the input values associated with Antarctic GIA, such as melting history since the Last Glacial Maximum and viscosity structure, based on the findings presented in this study. It is essential to consider how these uncertainties can be addressed through comparisons with forward modeling employed in prior research. If such matters are addressed elsewhere in the paper, it is advisable to provide a concise summary within this section. This recommendation also applies to the Ice Mass Change (IMC) discussion.

**Some typos:**

**P6, line 3:** We extent … -> extend?

**P9, line 16:** Although dominanted by the … -> dominated?

**P14, line 6:** IPCC Assessment report … -> IPCC Assessment Report?

**P14, line 17:** …but also prominant differences. -> prominent?

**P16, line 11:** …response times of millenia) -> millennia?

**P16, line 12** …deglaciation on millenial … -> millennial?

**P18, line 29** …$-144 \pm 27$ Gt a$-1$ und -> and?

---

## Author Response (AR1)

Once again we thank the three anonymous referees for their helpful and supportive comments on our manuscript. Please find below how we revised the manuscript in response to these comments. Italic font indicate the referees' comments. Green text indicates the authors' responses and we add marked-up text fragments indicating changes in the manuscript. Please find a complete marked-up manuscript version in a separate file, which highlights all changes made in the revised manuscript. Please also refer to our comments during the public discussion for the detailed responses to the referees' questions.

**Authors' response to Referee 1**

*According to section 2, the method builds upon the work from Rietbroek et al. (2016), which uses a variety of sea level fingerprints generated from mass change blocks or modes as basis functions to decompose the observations from GRACE and Altimetry. I would expect a paragraph describing the fingerprint dataset to be included in the manuscript for clarification.*

As proposed in our comment (https://doi.org/10.5194/tc-2023-119-AC1), we extended and added an explanation of the fingerprint inversion in Section 2.1 as follows:

The  global fingerprint inversion from Rietbroek et al. (2016) enables one to partition observed sea level, and to quantify the individual sea level budget components. For this purpose, globally consistent spatial patterns of the individual budget components are derived from a priori information. These spatial patterns serve as fingerprints in the inversion. Scaling factors for the individual fingerprints are then computed via a parameter estimation, utilizing observations from satellite altimetry over the ocean and satellite gravimetry. The quality of the a priori information crucially affects the final result. Rietbroek et al. (2016) found that the scaling factor of the Anatarctic GIA fingerprint in particular was estimated too low, meaning that the GIA effect determined in Antarctica is likely unrealistic.

*I am not sure if the inversion method in Rietbroek et al. (2016) works well when mass change blocks used to generate fingerprints are vertically superimposed. For example, GIA signals and ice mass change are overlapped, and their fingerprints are correlated. Therefore, they may not be well separated. If you are facing such a problem, please clarify how you have addressed it. In Rietbroek et al. (2016), I believe they initially removed an a priori GIA model. Please explain how you handled this problem in this study.*

As proposed in our comment (https://doi.org/10.5194/tc-2023-119-AC1), we clarified this as follows in Section 2.1:

 However, AIS IMC and GIA are superimposed in satellite gravimetry observations, i.e. a spatially resolved parametrization of these signals is strongly correlated and a signal separation appears challenging. For this reason, we additionally introduce satellite observations from ice-sheet altimetry over the AIS, which are sensitive to these signals as well. Furthermore, we make use of products from regional climate and firn modelling to account for ice-sheet surface processes.

*The input GRACE/GRACE-FO products are unfiltered spherical harmonic solutions complete to degree 96. We know that filtering techniques will introduce artefacts, but they also remove errors especially when the truncation degree is high. So, I was wondering if the authors have checked whether the filtering/smoothing will have a large impact on the final results. Looking at Figure 3, we see negative signals over DML and TA, which is not being found in other studies I think. Could this be caused by the unfiltered stripes? Also, why do not use Mascon solutions as input?*

Please refer to our comment (https://doi.org/10.5194/tc-2023-119-AC1) for a detailed answer to these questions. As proposed, we better clarified our strategy to account for errors in Section 2.1 by adding the following:

Here, our intentional goal is the incorporation of error-covariance information, which is a more rigorous approach to address the observational errors than minimizing error effects in the datasets by filtering. Furthermore, the large-scale fingerprints are not sensitive to small-scale errors, such as the typical GRACE/GRACE-FO stripe patterns.

In Section 2.2, we added an explanation for the choice of the used GRACE/GRACE-FO products:

It should be noted that GRACE/GRACE-FO level 3 products, e.g. mascon solutions, are not suitable for the investigation presented here due to the following reasons: (1) Mascon solutions are already corrected for the GIA effect, i.e. this GIA correction would have to be back-processed. (2) The globally consistent parametrization cannot applied to level 3 data and would have to be completely re-developed and (3) the rigorous propagation of covariance information would not be possible unless it is available along with the level 3 products.

*In Figure 3, the GIA results are compared with previous efforts. I am unsure if the GIA result in panel (a) contains the present-day GIA or not. The output GIA is compared with GNSS observations over Antarctica, so I assume it includes the present-day GIA effect. If it does contain present-day GIA, it may not be fair to compare it with the GIA from Caron et al. (2018)*

As proposed in our comment (https://doi.org/10.5194/tc-2023-119-AC1), we clarified what we mean by "present-day GIA effect" in Section 1 as follows:

The term present-day GIA effect refers to the presently observable effects resulting from the adjustment process to an isostatic state, which was induced by glacial mass changes in the past. This is to be distinguished from effects associated with the instantaneous elastic response to contemporaneous ice-mass loading changes (Thomas et al., 2011).

*Table 1 shows the AIS FAC result, but the signal appears to have a smaller value than the error. It's not clear if this has physical significance. Please provide clarification.*

We added the following notice in Section 3:

In view of the uncertainty, it is not possible to conclude whether the mean FAC rate for the entire grounded AIS is positive or negative during the 10-year time interval.

*P2 Line 7: what is the meaning of "...by subtracting the input and output mass fluxes"?*

We clarified this as follows:

and (iii) the mass budget method deriving the mass balance by  assessing the difference between the input and output mass fluxes.

*P4 Line 20: glacial isostatic adjustment → GIA*

Done.

*P6 Line 27: A GIA model is removed or not? To my knowledge, the conversion mentioned by Wahr et al. (1998) only applies to surface mass.*

We clarified this as follows:

We use the gravitational field changes ITSG-Grace2018 (Mayer-Gürr et al., 2018; Kvas et al., 2019), which are GRACE/GRACE-FO level-2 products provided as monthly sets of spherical harmonic coefficients up to degree  96 without any GIA correction.

*P9 Line 27: What is "SDS"?*

We added:

 Science Data System (SDS)

**Authors' response to Referee 2**

*In Section 4.3, the discussion primarily centers on the correlation between the spatial patterns derived through the current approach and the interplay between the analysis methodology and its outcomes. Furthermore, this section aims to enhance the physical comprehension of each dataset about Antarctica in this thesis. Critical objectives for this section include characterizing the spatial distribution of Antarctic Glacial Isostatic Adjustment (GIA) as determined by the current approach, summarizing the distinctive attributes and benefits of the present methodology, and offering physical interpretations. This entails a comparative analysis with forward modeling and other analytical techniques that have historically been utilized for separating Antarctic GIA and interpreting the physical processes of each result related to Antarctic mass balance.*

*Moreover, this section should touch upon the potential for constraining uncertainties within the input values associated with Antarctic GIA, such as melting history since the Last Glacial Maximum and viscosity structure, based on the findings presented in this study. It is essential to consider how these uncertainties can be addressed through comparisons with forward modeling employed in prior research. If such matters are addressed elsewhere in the paper, it is advisable to provide a concise summary within this section. This recommendation also applies to the Ice Mass Change (IMC) discussion.*

As proposed in our comment (https://doi.org/10.5194/tc-2023-119-AC2), we revised the structure and content of the interpretation of our results presented in Section 4.3 as follows:

[revised manuscript text omitted]

*P6, line 3: We extent ... → extend?*

Done.

*P9, line 16: Although dominanted by the ... → dominated?*

Done.

*P14, line 6: IPCC Assessment report ... → IPCC Assessment Report?*

Done.

*P14, line 17: ... but also prominant differences. → prominent?*

Done.

*P16, line 11: ... response times of millenia) → millennia?*

Done.

*P16, line 12 ... deglaciation on millenial ... → millennial?*

Done.

*P18, line 29 ... −144 ± 27 Gt a⁻¹ und → and?*

Done.

**Authors' response to Referee 3**

*1) I find the split of figures between then main paper and the supplementary information imbalanced. There are only 3 figures in the main body of the paper but 16 contained in the supplementary information. I think this manuscript could be improved by moving some figures to the main body, such as one figure for each assessment method detailed in Section 2.3. E.g. S10 for (1), Fig S9 or S13 for assessment (3). Furthermore, each assessment is clearly set out in section 2.3 and described in separate paragraphs. In Section 3, it would be useful to describe the results of each assessment by referring to the number (1,2,3) as described in Section 2.3*

As proposed in our comment (https://doi.org/10.5194/tc-2023-119-AC3), we moved Figure S9 and S13 from the Supplement to the main body. These are now Figure 3 and 5. Furthermore we applied the way of numbering the assessment methods, introduced in Section 2.3, to Section 3 and to Figures 3–5.

*2) I have a slight issue with the treatment of the Antarctic Peninsula as described in the supplementary information. As the authors point out, the expected mantle viscosity here is low (see also Nield et al. (2014), Samrat et al. (2021)). As such, using the ICE6G loading model, which neglects any ice loading in the Late Holocene, will produce incorrect results – since the Late Holocene ice mass changes will dominate the present-day signal. Combined with SELEN, which is likely not high resolution enough to capture GIA here, I think this limitation in the method should be mentioned in the main text to make it clear this area is from a forward model (Pg 5, Line6-7). What was the reason for not using the Caron forward model of GIA in the Peninsula?*

As proposed in our comment (https://doi.org/10.5194/tc-2023-119-AC3), we moved the reasoning for the adapted GIA-parametrization on the Antarctic Peninsula from the Supplement to Sect. 2.1. We made the following changes:

An exception from forward-model independent GIA parametrization is made on the Antarctic Peninsula. From our validation experiments, we found that we were not able to retrieve reasonable GIA results for the northern part of the Antarctic Peninsula (Graham Land). We attribute this mainly to the insufficient quality of surface elevation changes derived from radar altimetry here (e.g. Schröder et al., 2019). In turn the significant misfit between GRACE/GRACE-FO products and CryoSat-2 products is captured by an unphysical GIA signal. This is also the case for other inverse GIA estimates (e.g. Gunter et al., 2014; Engels et al., 2018; Willen et al., 2020). To prevent an unphysical GIA, we decided not to co-estimate GIA  in this particular region. We did not include local GIA patterns on the Peninsula in our local GIA-pattern parametrization. Instead we approach the GIA effect here by a global GIA model result which is then subtracted from the observations. We model the GIA effect with an ICE-6G ice history that solely exists in the Graham Land Region by using SELEN[4] (Spada and Melini, 2019). Figure S1 illustrates the modified GIA parametrization with the Antarctic Peninsula GIA pattern. Admittedly, this GIA pattern has strong limitations to represent the true GIA effect in this region. The upper-mantle viscosity is found to be low here (Nield et al., 2014; Samrat et al., 2021; Ivins et al., 2021). We therefore expect that GIA response time scales are similar to those in the Amundsen Sea Embayment region. This means that the applied pattern (Fig. S1) will only allow an incomplete representation of the actual GIA and will not resolve GIA effects induced by load changes over the last centuries. Nevertheless, we argue that this methodological adjustment allows to, at least, limit the bias to the entire Antarctic GIA estimate.

Furthermore, we added to Sect. 4.4 the following:

We expect a significant quality improvement of satellite altimetry-derived surface elevation changes with new retracking methods in case of radar altimetry (Helm et al., 2023) and with the growing availability over time of laser altimetry products from the ICESat-2 mission. In terms of mean rates, the quality of the results in general will grow by investigating longer time periods. For instance, it is expected that the enhanced data products will facilitate the observation-based assessment of the GIA effect on the Antarctic Peninsula, which was not achieved in this study.

*P1, Line3: Define GRACE on first use.*

Done.

*P1, Line 12: Define and reference IMBIE.*

Done.

*P2, Line 19: "hardly characterized uncertainties" not sure what this means, consider rephrasing.*

We clarified this as follows:

(ii) has the advantage to capture IMC with high spatial resolution (e.g., Schröder et al., 2019) but the conversion from volume changes to mass changes is based on effective density hypotheses or needs to include auxiliary data, e.g. firn modelling results  where the uncertainties are largely unknown.

*P3, Line 11: "This GIA parametrization allows to spatially resolve the GIA effect in Antarctica unpredicted by GIA forward modelling" What do you mean by "unpredicted"? Without relying on GIA forward modelling, or that resolving GIA in this way is revealing something that is not predicted by current GIA forward models?*

Here we mean the latter and clarified this as follows:

*P3, Line 17: can you explain why you limit to this particular 10 year period?*

We added the following explanation:

We present and analyse results from applying this approach over the 10-year observation period from Jan 2011 to Dec 2020 (2011–2021)  during which the following data sets are available at the same time: a satellite gravimetry data product from GRACE and GRACE-FO (ITSG-Grace2018 Mayer-Gürr et al., 2018), a satellite altimetry data product from CryoSat-2 (Helm et al., 2014), and changes of FAC derived from RACMO2.3p2 SMB (van Wessem et al., 2018) and the IMAU-FDMv1.2A (Veldhuijsen et al., 2023). We validate the results with independent GNSS data.

*P5, Lines 3-5: The impulse response patterns are generated with the code SELEN, which is a forward model. In what way does it allow "capture GIA effects independent from GIA forward models"? Consider clarifying here.*

We clarified this as follows:

 In principle, this GIA parametrization allows to spatially resolve  GIA effects in Antarctica, which have not been predicted by GIA forward modelling.

*P10, line 12 "Not" → Note*

Done.

*P19, Line 2 "than GIA from others" → "GIA from other inversion studies"?*

Done.